# Attentional Bias Is Associated with Negative Emotions in Problematic Users of Social Media as Measured by a Dot-Probe Task

**DOI:** 10.3390/ijerph192416938

**Published:** 2022-12-16

**Authors:** Jin Zhao, Zinuan Zhou, Bo Sun, Xinyuan Zhang, Lin Zhang, Shimin Fu

**Affiliations:** 1School of Education, Guangzhou University, Guangzhou 511400, China; 2Dongguan Songshan Lake Experimental Middle School, Dongguan 523000, China; 3School of New Media, Financial & Economic News, Guangdong University of Finance, Guangzhou 510521, China

**Keywords:** problematic social media use, negative emotion, attentional bias, Stroop task, dot-probe task

## Abstract

Objective: Social media (SM) have flourished and are affecting human lives on an unprecedented scale. Problematic social media use (PSMU) is a recently emerging problematic behavior that affects both physical and mental health. The purpose of this study was to investigate whether problematic users of SM display attentional bias (AB) toward SM-related cues, as well as the relationships between AB, the severity of PSMU, and negative emotions. Method: 1000 college students were recruited through classes and online forums and then tested using the Bergen Social Media Addiction Scale (BSMAS). Eventually, 60 participants, identified by the cut-off point for BSMAS, consented to participate in the study and were divided into two groups (i.e., a problematic use group and a typical use group). The severity of PSMU and negative emotions (i.e., anxiety, depression, social fear, and loneliness) were evaluated by self-report questionnaires. AB was assessed by an addiction Stroop task and a dot-probe task (DPT). Results: PSMU was found to be positively associated with anxiety (r = 0.28, *p* < 0.05), depression (r = 0.35, *p* < 0.01), and social fear (r = 0.38, *p* < 0.01), but not with loneliness (r = 0.19, *p* = 0.15). Participants with a tendency toward PSMU displayed AB toward SM-related cues in the DPT [F (1, 58) = 26.77, *p* < 0.001, $\eta_{p}^{2}$ = 0.32], but not in the Stroop task [F (1, 58) = 0.61, *p* = 0.44, $\eta_{p}^{2}$ = 0.01]. Moreover, AB toward SM-related stimuli was found to be positively correlated with the severity of PSMU (r = −0.51, *p* < 0.001), anxiety (r = −0.37, *p* < 0.01), depression (r = −0.51, *p* < 0.001), and social fear (r = −0.30, *p* < 0.05) in the DPT. Conclusions: Problematic users of SM show AB towards SM-related cues in the DPT, which is more reliable for assessing AB than the Stroop task. Moreover, it is suggested that in clinical interventions we should work to change the AB towards SM-related stimuli and improve negative emotions to decrease risks of PSMU.

## 1. Introduction

With the popularity of the Internet, social media (SM) (e.g., WeChat, TikTok, Facebook) have flourished and are affecting human lives on an unprecedented scale. People use SM to experience and engage in various types of social activities and entertainment, such as online social networking, playing games, sharing pictures, and passing time. SM not only provide great convenience for people to establish and expand their social relations, but also provide an important window through which people can understand external information [1]. According to a 2021 Pew Research Center report, YouTube and Facebook dominate SM landscape in the U.S., attracting 81% and 69% of the adult population, respectively. Seven-in-ten Facebook users report that they use the site every day, including 49% of users who log in to the site several times a day. However, irrational use of SM may result in problematic social media use (PSMU), which refers to the compulsive and excessive use of SM platforms and networking sites, leading to a severe impact on all aspects of life [2]. A recent survey reported a potential prevalence rate for PSMU of 3.5% in a Chinese sample [3]. In recent years, there has been a growing interest in studies on the consequences of PSMU. For example, De Doncker and McLean found that PSMU is positively correlated with poor sleep quality and depressive symptoms [4]. Zhao revealed that compared with typical users of SM, problematic users of SM have lower subjective well-being [5]. In addition, Ali Homaid found that PSMU induced both exhaustion and technostress, leading to decreased academic performance [6].

In the literature on PSMU, there has been considerable debate regarding whether PSMU should be regarded as a genuine addiction behavior. Although PSMU is not formally recognized in the current psychiatric nosology, a growing body of research has found that PSMU is similar to substance and behavioral addictions, exhibiting six core symptoms of addictions [7,8]: salience (time and energy spent in SM neglecting other activities), mood modification (using SM to experience a positive mood state and/or escape negative emotions and feelings of stress), tolerance (need to increase the intensity of SM use to achieve the same mood effect), withdrawal (unpleasant feeling states and behavioral discomforts when the use of SM is discontinued or suddenly reduced), conflict (interpersonal and intrapsychic conflicts due to difficulty limiting the time and frequency of using SM), and relapse (a tendency to try to reduce or stop using SM, but failing, and thus returning to SM).

### 1.1. Negative Emotions and PSMU

Negative emotions are potential risk factors for PSMU. The theory of an affective processing model of negative reinforcement explains the mechanism of the formation, maintenance, and development of addictive behaviors from the perspective of emotional processing [9]. This theory suggests that negative emotions are the dominant motivation to maintain substance addiction and behavioral addictions. Individuals are more likely to form addictive behaviors when they wish to escape from negative emotions. Numerous studies have investigated the influences of anxiety, depression, social fear, and loneliness on PSMU. Severe or even moderate anxiety and/or depression may increase risks of developing PSMU [10]. Compared with non-users of the SM platform, people who use Facebook more often tend to suffer from insomnia and depression [11]. In addition, individuals with high levels of social fear and loneliness are more likely to turn to SM as a way to avoid uncomfortable face-to-face interactions and compensate for the lack of offline relationships [12]. For example, de Bérail et al. adopted the structural equation modeling to explore the relationship between social fear and PSMU. They found social fear increased risks of YouTube addiction, which directly and indirectly influenced addictive behavior [13]. In addition, a study by Primark et al. indicated that loneliness is one of the risk factors for PSMU [14]. To briefly summarize, these results suggest that negative emotions may partially contribute to the development of PSMU.

### 1.2. Attentional Bias towards Addiction-Related Cues in Addictions

The contemporary conceptualization of addictive behaviors is not limited to substance addiction (e.g., nicotine addiction, alcohol abuse, and heroin dependence). Increasing research attention has been paid to various forms of behavioral addiction (e.g., online shopping addiction, pathological gambling, and online pornography). Substance and behavioral addictions are both thought to exhibit the same underlying biological psychosocial addiction mechanism [15]. It is well-established that addicts tend to direct their attention to addiction-relevant cues in the form of attentional bias (AB), a strong tendency to prioritize the allocation of attentional resources to addiction-related stimuli [16].

Researchers have adopted multiple paradigms to study AB in substance and behavioral addicts, including both indirect and direct measures. The addiction Stroop task, a modified version of the classic Stroop task, is an indirect measure most widely used to investigate AB. The presence of AB can be inferred if participants’ performance on the color-naming task is impaired when addiction-relevant cues are presented simultaneously [17]. In general, a longer response time (RT) in the presence of addiction-related stimuli suggests AB to addiction-related cues. The dot-probe task (DPT) is another commonly used indirect measure of how addiction-related cues capture participants’ visuospatial attention. In the DPT, a pair of stimuli consisting of an addiction-relevant stimulus (e.g., a nicotine-related picture) and a matched control stimulus (e.g., a landscape picture), is presented on the screen simultaneously. When the pair of stimuli disappears, a probe stimulus appears at the location previously occupied by either the addiction-relevant stimulus or the control stimulus. The presence of AB is inferred from the comparison of participants’ RT to the probe stimuli that replace addiction-related cues vs. those that replace the control cues [18]. If the participants respond faster to a probe stimulus at the location of an addiction-related stimulus than at the location of a neutral stimulus, one can infer AB to addiction-related cues. Other less widely used indirect measurements include flicker-induced change blindness paradigms, dual-task procedures, and visual search paradigms [19]. Eye-tracking technology, as a method of directly measuring AB, allows for the measurement of individuals’ visuospatial selective attention when addiction-related stimuli are presented.

Numerous studies employing a range of experimental paradigms have revealed that there is a correlation between addiction and AB towards addiction-related cues [20,21,22]. Currently, the most influential theoretical explanation of AB in substance addicts is the incentive–sensitization theory [23]. According to this theory, addictive behaviors are persistent and progressive neuroadaptations caused by repeated substance use. The repeated exposure to highly pleasurable cues leads to hypersensitization of the reward systems in the brain that generate incentive salience toward addiction-related cues. Incentive sensitization produces both a bias of attentional processing towards substance-relevant stimuli and pathological motivations for addictive stimuli: in other words, a principle of “grab attention, become attractive and wanted, and guide behavior to the incentive”. The theory argues that “wanting” (strong motivation) drives compulsive behaviors in substance use more than “liking” (hedonic pleasure). The incentive–sensitization theory was initially used only to explain substance addiction. Further studies have demonstrated that the theory can also explain behavioral addictions. Overall, the role of AB towards addiction-related cues is purported to play a significant role in the development, maintenance, and relapse stages of both substance and behavioral addictions [24].

PSMU differs from most substance and behavioral addictions in that the availability of SM is so high that everyone has the opportunity to easily access it. Individuals can use SM on different platforms for different activities. Although AB towards addiction-related stimuli has been extensively studied in substance and behavioral addicts, it is not clear whether SM-related cues have similar motivational properties to capture and maintain the attention of SM users. To the best of our knowledge, only two studies have explored AB in problematic users of SM. However, their findings are inconsistent. Nikolaidou et al. used eye-tracking technology in combination with a DPT to assess AB and found that problematic users of SM tend to show increased attentional dwell times toward SM-related stimuli [25]. Their findings support the hypothesis that the AB effect is a common underlying mechanism related to PSMU and other addictions. However, another study by Thomson et al. did not provide support for the presence of the AB effect in PSMU [26]. Their study found that problematic users of SM showed no preferential attentional capture by SM apps and notifications with a visual search paradigm. Furthermore, the engagement and severity of PSMU were not found to be correlated with the AB effect. The main reason for the different results may be the different paradigms used in the two studies. Hence, the issue of AB towards SM-related stimuli in problematic users of SM needs to be explored further.

### 1.3. Links between AB towards Addiction-Related Cues and Negative Emotions

The experience of negative mood is thought to be a reliable trigger of PSMU. According to the theory of an affective processing model of negative reinforcement, negative emotions are the core symptoms of addiction withdrawal syndrome. Negative emotions can disrupt an individual’s homeostasis, which has been considered the key to the pathophysiological maintenance of addiction, and enhance an individual’s motivation to form and maintain addictive behaviors such as alcohol abuse, nicotine dependence, and other forms of addictions [9]. Therefore, it is reasonable to hypothesize that negative emotions increase AB towards addiction-relevant cues in addicts and some evidence supports this hypothesis. For example, Field and Powell used the DPT and found that alcoholics showed increased AB towards alcohol-related cues after stress and anxiety were induced, manifesting in both the initial orientation of attention and attentional retention [27]. For problematic users of SM, under negative emotions individuals actively seek SM-related cues and direct their attention to the previous pleasurable events in order to escape from unhappiness and improve their emotional states. Therefore, we can speculate that problematic users of SM preferentially process SM-related cues and use online chatting, games, shopping and other functions to alleviate negative emotions. To the best of our knowledge, no study has revealed any links between AB towards SM-related cues in problematic users of SM and their negative emotions.

### 1.4. Purposes of the Present Study

Prior studies analyzing the relationships between AB, severity of PSMU, and negative emotions have been rare. Therefore, the current study had three goals. First, we explored the associations of negative emotions (i.e., anxiety, depression, social fear, and loneliness) with PSMU using self-reported questionnaires. Second, an attempt was made using an addiction Stroop task and a DPT to examine whether SM addicts show AB towards SM-related stimuli. Both tasks were included to improve the reliability of the current study. We also tested whether the severity of PSMU is associated with AB towards SM-related cues. Third, the link between AB and negative emotions was analyzed. It was hypothesized that:

**Hypothesis 1 (H1):** 
*Negative emotions (i.e., anxiety, depression, social fear, and loneliness) are positively correlated with PSMU.*


**Hypothesis 2 (H2):** 
*SM addicts show greater AB towards SM-related cues and the severity of PSMU is positively correlated with AB toward SM-related stimuli.*


**Hypothesis 3 (H3):** 
*Increased AB toward SM-related cues is positively correlated with higher levels of negative emotions.*


## 2. Materials and Methods

### 2.1. Participants

The size of the necessary sample was calculated a priori using G*Power. A total sample size of 36 participants is needed for α = 0.05, f = 0.25, and a power of 0.95. Following the example of Hu et al. [28], in order to improve the internal reliability of this study, 30 participants were recruited for the problematic use group and typical use group, respectively. Considering that the prevalence of PSMU is approximately 3.5% [3], 1000 college students were recruited through classes and online forums at Guangzhou University and then tested using the Bergen Social Media Addiction Scale (BSMAS). Inclusion criteria included fluency in Chinese, normal or corrected-to-normal vision, and the use of SM. Exclusion criteria included current or previous psychiatric conditions (schizophrenia, mood disorders, obsessive compulsive disorder, and somatoform disorders), and addiction behaviors including gaming addiction, alcohol problems, and nicotine dependence. Structured clinical interviews were used to determine whether participants met the inclusion and exclusion criteria. According to the previous study, the cut-off point for BSMAS was 24 points [3]: if the BSMAS score was 24 or above, the participant was classified into the “problematic use group”. Otherwise, the participant was classified into the “typical use group”. Ultimately, 60 participants, including 30 in the problematic use group and 30 in the typical use group, consented to participate in the experiment. All participants received CNY 15 for their participation.

### 2.2. Procedure

Participants were asked to complete the questionnaires, an addiction Stroop task, and a DPT in a quiet room. There was a 2 min break between each of the two tasks, and the order of the addiction Stroop task and the DPT was balanced.

### 2.3. Instruments

BSMAS: The Chinese version of the BSMAS [29], adapted from Andreassen et al. [30], was adopted to measure the severity of PSMU. It comprises 6 items (e.g., “How often have you felt an urge to use SM during the last year?”) measured on a 5-point Likert scale (from 1 = *very rarely* to 5 = *very often*) with higher scores indicating greater levels of PSMU. The Cronbach’s alpha coefficient for the BSMAS was 0.91 in the present study.

Generalized Anxiety Disorder-7 (GAD-7): The Chinese version of the GAD-7 [31], adapted from Spitzer et al. [32], was used to measure anxiety. It comprises 7 items (e.g., “I feel restless, anxious, and irritable”) measured on a 4-point Likert scale (from 0 = *not at all* to 3 = *nearly every day*) with higher scores indicating greater levels of anxiety. The Cronbach’s alpha coefficient for the GAD-7 scale was 0.92 in the present study.

Patient Health Questionnaire-9 (PHQ-9): The Chinese version of the PQH-9 [33], adapted from Kroenke et al. [34], was used to measure depression. It comprises 9 items (e.g., “I feel down, depressed or hopeless”) measured on a 4-point Likert scale (from 0 = *not at all* to 3 = *nearly every day*) with higher scores indicating greater levels of depression. The Cronbach’s alpha coefficient for the PHQ-9 scale was 0.79 in the present study.

Anxiety Scale (IAS): The Chinese version of the IAS [35], adapted from Leary [36], was used to measure social fear. It comprises 15 items (e.g., “I will be nervous during an interview”) measured on a 5-point scale (from 1 = *not at all consistent* to 5 = *extremely consistent*) with higher scores indicating greater levels of social fear. The Cronbach’s alpha coefficient for this scale was 0.89 in the present study.

UCLA Loneliness Scale: The Chinese version of this scale [37]., adapted from Russell [38], was adopted to measure loneliness. It comprises 20 items (e.g., “Do you often feel that no one can be trusted?”) measured on a 4-point scale (from 1 = *never* to 4 = *always*) with higher scores indicating greater levels of loneliness. The Cronbach’s alpha coefficient for this scale was 0.93 in the present study.

Addiction Stroop task: Before the task, 15 college students who did not participate in this experiment were asked to rate the extent of SM-related words and neutral words on a 5-point Likert scale (from 1 = *extremely inconsistent* to 5 = *extremely consistent*). Ultimately, 20 SM-related words (e.g., WeChat, Weibo, circle of friends) and 20 neutral words referring to daily necessities (e.g., desk, glasses, and shampoo) were chosen for use in the formal experiment. SM-related words and neutral words were matched in similarity, frequency, and length.

The experimental program was written and ran on E-Prime. The stimuli were presented on a 27-inch LCD monitor with 1024 × 768 pixels resolution and a 60 Hz refresh rate. During the task, a fixation cross (+) with a duration of 500 milliseconds (ms) was first presented at a 60 cm viewing distance. After the fixation cross disappeared, a stimulus in Arial, font size 48, was displayed in the center of the screen. Words were presented sequentially in a block, in randomized order, with their color either blue or red, resulting in 80 stimuli for each block. The task comprised 3 blocks and a total of 240 trials. Participants were required to recognize the color of each stimulus as accurately and quickly as possible, while ignoring the actual meaning of the stimulus. If the word was red, the participant was required to press the “F” key. If the word was blue, the participant was required to press the “J” key. Each word remained onscreen until either a key was pressed or 3000 ms had elapsed (Figure 1). No color or word was presented more than twice consecutively. There was a 30 sec break between each block and the next block was initiated by the participant. Before the formal experiment, participants performed 20 practice trials with feedback to ensure that they understood the requirements and procedures. The stimuli used in the practice were not reused in the formal experiment.

Addiction DPT: Stimuli for the addiction DPT were selected as described for the addiction Stroop task. The timeline of the addiction DPT is shown in Figure 2. During the task, a fixation cross “+” was presented first at the center of the screen for a duration of 500 ms. It was followed by a 1000 ms presentation of a pair of words (one SM-related word and one neutral word) located on the left and right sides of the cross. A blank was presented for 50 ms, followed by a probe dot. Participants were asked to determine the location of the probe dot: they were instructed to press the “F” or “J” key if the probe appeared on the left or right side of the screen, respectively. The probe remained on the screen until either a key was pressed or 2000 ms had elapsed. Finally, a blank screen was presented for 500 ms and the next trial was performed. As in the addiction Stroop task, participants were first provided 20 practice trials with feedback to ensure that they understood the procedure. Next, each participant was shown 20 pairs × 2 probe positions × 2 picture positions (i.e., 80 trials) in a fully randomized order in one block with no feedback. Participants performed a total of 240 trials, which were divided into three blocks with a 30 ms interval between each. Two types of trials were used in the paradigm, e.g., congruent (the probe appeared where the SM-related words had appeared) and incongruent (the probe appeared where the neutral words had appeared). Stimuli in Arial, font size 48, were presented on both sides of the screen. The center of the probe dot was 12 cm from the center of the screen.

## 3. Results

### 3.1. Demographics

The two groups of participants did not differ with regard to gender [*χ^2^* (1) = 0.32, *p* = 0.57] or age [t (58) = 1.83, *p* = 0.07]. The scores for the BSMAS in problematic use group were significantly higher than those in the typical use group [t (58) = 10.29, *p <* 0.01]. See Table 1 for details.

### 3.2. Questionnaire Data: Inter-Correlations between Variables

Bivariate correlations of variables are presented in Table 2. It can be seen that anxiety, depression, and social fear were significantly positively correlated with BSMAS (r = 0.27, *p <* 0.05 for anxiety; r = 0.35, *p <* 0.01 for depression; r = 0.42, *p <* 0.01 for social fear), while loneliness was not significantly associated with BSMAS (r = 0.19, *p =* 0.15).

### 3.3. Behavioral Experiment Data: Analyses of AB

Analysis of errors: The data were screened as follows: first, false trials were removed; then, extreme responses defined as extreme values outside of 200–1200 ms were removed; and finally, data that were three standard deviations from the mean were removed.

Addiction Stroop task: A 2 × 2 ANOVA of trial type (SM-related words, neutral words) × group (problematic use group, typical use group) was conducted with trial type as the within-subject factor and group as the between-subject factor. There was no main effect for trial type [F (1, 58) = 2.65, *p* = 0.11, $\eta_{p}^{2}$ = 0.04], or for group [F (1, 58) = 0.05, *p* = 0.82, $\eta_{p}^{2}\;$= 0.001]. No significant interaction effect for group × trial type was found [F (1, 58) = 0.61, *p* = 0.44, $\eta_{p}^{2}\;$= 0.01]. This indicates that problematic users of SM displayed no special AB toward SM-related words in the addiction Stroop task. See Figure 3A for details.

Addiction DPT: A 2 × 2 ANOVA of trial type (congruent, incongruent) × group (problematic use group, typical use group) was conducted with trial type as the within-subject factor and group as the between-subject factor. There was a main effect for trial type [F (1, 58) = 14.77, *p <* 0.001, $\eta_{p}^{2}$ = 0.20]. RT in the congruent condition was less than that in the incongruent condition. There was no main effect for group [F (1, 58) = 3.03, *p* = 0.09, $\eta_{p}^{2}$ = 0.05]. An interaction was found for group × word type [F (1, 58) = 26.77, *p <* 0.001, $\eta_{p}^{2}$ = 0.32]. LSD post-hoc tests revealed that the problematic users of SM reacted faster to the congruent condition than to the incongruent condition [*t* (29) = −4.64, *p <* 0.001, −20 ms]. Typical users of SM reacted more slowly to the congruent condition than to the incongruent condition [*t* (29) = 2.86, *p <* 0.01, 2 ms]. The problematic use group responded more quickly to the congruent trials than the typical use group [*t* (58) = −3.00, *p* = 0.04]. There was no significant difference in the mean RT of the two groups toward the incongruent trials [*t* (58) = 0.56, *p* = 0.58]. This suggests that problematic users of SM displayed AB toward SM-related words and that they responded faster to SM-related stimuli than typical users in the addiction DPT. See Figure 3B for details.

### 3.4. Relationship between Questionnaire Data and the Index of AB

Correlation between the severity of PSMU and the index of AB: There was no significant association between PSMU and the index of AB in the addiction Stroop task (r = 0.23, *p* = 0.08). PSMU was found to be negatively correlated with the index of AB in the DPT (r = −0.51, *p <* 0.001). See Figure 4 for details.

Links between negative emotions and the index of AB: Correlational analysis was conducted between the index of AB and negative emotions to determine any factor(s) that could affect the index of AB toward SM-related stimuli. There was no significant correlation between negative emotions (i.e., anxiety, depression, social fear, and loneliness) and the index of AB in the addiction Stroop task (r = 0.16, *p =* 0.21 for anxiety; r = 0.12, *p* = 0.37 for depression; r = 0.25, *p =* 0.06 for social fear; r = 0.15, *p =* 0.26 for loneliness). The index of AB was significantly negatively correlated with anxiety, depression, and social fear in the DPT (r = −0.37, *p <* 0.01 for anxiety; r = −0.51, *p <* 0.001 for depression; r = −0.30, *p <* 0.05 for social fear), but not with loneliness (r = −0.20, *p =* 0.13). See Figure 5 for details.

## 4. Discussion

In the current study, we used self-report scales to look at the relationship between PSMU and negative emotions. Anxiety, depression, and social fear were found to be positively associated with PSMU, but loneliness was not. We also used the two most commonly utilized experimental paradigms to examine whether problematic users of SM showed AB towards SM-related stimuli. An attempt was also made to explore the relationship between the severity of PSMU and AB. Compared with typical users, problematic participants showed greater AB towards SM-related stimuli, and the AB toward SM-related information was positively associated with the severity of PSMU in the DPT, not but in the addiction Stroop task. Finally, we analyzed the relationship between AB and negative emotions. Our key findings were that AB was positively correlated with anxiety, depression, and social fear in the DPT. As far as we know, the links between negative emotions and AB directed at SM-related stimuli have rarely been studied in the prior literature.

### 4.1. Relationships between Negative Emotions and PSMU

Anxiety, depression, and social fear were found to be positively associated with PSMU. PSMU has generally been considered to be a coping strategy for dealing with anxiety and/or depression [39]. A meta-analysis has confirmed the positive relationship between Facebook use and depression [40]. In the present study, social fear was found to be positively associated with PSMU. This is consistent with the study by Zhao, which suggested that social fear is a potential risk factor for PSMU [12]. People who have difficulty expressing themselves in face-to-face social environments prefer to express themselves on SM, where they can express themselves more easily and perceive less risk. Interestingly, this study found the correlation between loneliness and PSMU to be relatively low. This is inconsistent with previous studies that have indicated that loneliness may increase the risk of PSMU [14,41]. It has been postulated that a lack of correlation between loneliness and PSMU may attributable to the fact that more lonely individuals experience greater stress while using SM than less lonely individuals, limiting their ability to realize the potential benefits of PSMU [42].

### 4.2. AB towards SM-Related Cues in Problematic Users of SM and Links between PSMU and AB

This study used two experimental tasks to explore whether problematic users of SM showed AB towards SM-related cues. The problematic use group was found to display a specific AB toward SM-related stimuli in the addiction DPT. This result supports our hypothesis that SM addicts have a greater AB towards SM-related stimuli. AB towards addiction-related cues is considered to account for the maintenance of the addiction cycle [20,25]. Nikolaidou et al. previously found that problematic users of SM tend to show an increased AB towards SM-related stimuli [25], but contrarily, Thomson et al. found that there was no special AB effect towards SM-related cues in problematic users [26]. The current study was consistent with the former finding. There have been many studies supporting the claim that AB towards addiction-related information plays an important role in addiction behavior. For example, Wilcockson et al. [43] used a behavioral inhibition task and a DPT to find that compared with the non-smokers, dependent smokers showed decreased inhibition and increased AB for smoking-related cues. Similarly, Kim et al. found that people with online gaming disorder showed AB towards game-related stimuli using eye-tracking technology and their study supported that AB towards addiction-relevant cues was one of the core symptoms of addictive behaviors [44]. The current study found that SM-related information can capture the attention of problematic users of SM; thus, we are more inclined to suggest that PSMU is an addictive behavior and thus exhibits similar addiction mechanisms to behavioral addiction. Additionally, in agreement with studies that have shown that the AB effect may indicate the severity of the addiction [45], our study found a negative correlation between the severity of PSMU and the index of AB towards SM-related cues in the DPT. In other words, the more serious individuals’ PSMU, the more attention resources they devote to SM-related information. This result suggests that AB toward SM-related cues may be a reliable indicator of the severity of PSMU.

Interestingly, in the addiction Stroop task, we found neither an AB effect in SM addicts, nor a significant relationship between severity of PSMU and AB. Contrary to our current results, Jeromin et al. used a Stoop task and a DPT to study the AB of excessive Internet gamers, and found that in the Stroop task, excessive Internet gamers responded more slowly to computer-related words than to neutral words, but in the DPT, there was no difference between targets following neutral and computer-related images [46]. This discrepancy may be because of the following reasons: First, stimuli were presented on the screen side by side in the DPT, while all stimuli were presented at the center of the screen in the Stroop task. When using words as stimulus material, the DPT is a more sensitive tool for assessing AB than the Stroop task. Second, the relatively long duration of stimulus presentation (3000 ms) in the Stroop task may have allowed participants to consciously control their attentional resources. Garland et al. used an addiction Stroop task with a 100 ms inter-stimulus interval and found that an automatic AB for substance-related stimuli plays a major role in provoking cravings in alcohol-dependent individuals [47]. Over time, AB towards addiction-related cues becomes automatic and implicit. Third, the Stroop task may have poor stability and reliability. Several studies have failed to find an association between addiction and AB using the addiction Stroop task [48,49]. Last, a possible explanation for this inconsistency could be that the Stroop task may be too easy for problematic users of SM. Hence, it can be concluded that the DPT paradigm is a more reliable measurement for evaluating AB than the addiction Stroop task, as found in the current study.

### 4.3. Links between AB towards SM-Related Information and Negative Emotions

This was the first study to look at the relationship between AB towards SM-related cues and negative emotions. In the DPT, AB was found to be positively associated with negative emotions, especially anxiety, depression, and social fear. This result can be explained by the theory of an affective processing model of negative reinforcement [9]. According to this theory, negative emotions are an important factor in maintaining addictive behaviors. In order to escape from unpleasant feelings and improve their emotional level, addicts actively seek addictive cues or form addictive behaviors. This study supports the findings of Field and Powell [27], which showed that negative emotions can increase the AB of addicts toward addiction-related cues.

Interestingly, this study found that AB was positively correlated with anxiety, depression, and social fear, but had a relatively low correlation with loneliness. According to the theory of the antecedent model of Internet addiction and psychological adaptation, Internet addiction can lead to psychological maladaptation. Prior research has found that addiction and depression often occur together. Addiction may increase the risk of depression and is an important predictor of depression [50]. A study by Ciarrochi et al., confirmed through a four-year study of adolescents that Internet addiction is a consistent factor leading to depression [51]. Problematic users of SM have a higher probability of suffering from major depressive disorder. PSMU has been found to be positively correlated with depression and is considered to be a significant risk factor in the occurrence and development of depression [52]. As this study found, significant AB often predicts severe PSMU. Based on the studies described above, we can understand that there is a positive correlation between AB and depression. Anxiety is generally considered to be highly correlated with depression. Jacobson and Newman explored the relationship between early anxiety and later depression using a longitudinal tracking approach and found that anxiety predicts later depression in structural equation models [53]. Therefore, we can understand the close connection between anxiety and AB toward SM-related cues. In addition, there is a significant positive correlation between social fear and PSMU, as found in the present study. With an increase in individuals’ dependence on SM, they spend more time and energy on SM, and less time and energy on interpersonal communication and other activities, thus weakening their sense of self-efficacy, enhancing their sense of loneliness, and ultimately leading to social fear [50]. Therefore, we can conclude that individuals’ greater AB towards SM-related cues can increase their risk of social fear. The relatively weak correlation between loneliness and AB towards SM-related cues is mainly due to the greater stress experienced by lonely individuals, limiting their ability to realize the potential benefits of PSMU [42]. Therefore, SM-related stimuli fail to capture their attention.

## 5. Conclusions

Our research not only contributes to our understanding of the role of AB in the development and maintenance of PSMU, but also has important implications for interventions and nursing against PSMU. PSMU was found to be positively correlated with anxiety, depression, and social fear. Individuals with PSMU showed significant AB toward SM-related cues in the DPT paradigm, but not in the addiction Stroop task. Moreover, AB toward SM-related stimuli was positively correlated with the severity of PSMU, anxiety, depression, and social fear in the DPT paradigm. Future interventions for SM addicts may start with correcting their AB with AB modification programs to decrease the risk of PSMU. Supporting evidence comes from the study of Kakoschke et al. to discourage the consumption of unhealthy food [54]. In the ‘attend healthy’ group, 90% of the dot probes were at the locations of healthy foods, while in the ‘attend unhealthy’ group, 90% of the dot probes were at the locations of unhealthy foods. Their study found that participants in the ‘attend healthy’ group showed an increased AB toward healthy foods-relevant cues and ate relatively more healthy snacks than the participants in the ‘attend unhealthy’ group.

## 6. Limitations and Future Directions

The present study has some limitations. First, only BSMAS was used in diagnosing PSMU. Because PSMU is a controversial diagnosis, whether a single scale can accurately distinguish and identify such behaviors remains debatable. Second, in the Stroop task, the presentation time of the stimuli was too long, which may have affected the reliability of the Stroop task. Third, word stimuli were used in both the Stroop task and the DPT as experimental materials, and the ecological effect of text may not be as good as that of pictures. The conflicting findings of this study indicate that other research paradigms are needed to explore AB in PSMU. Fourth, the sample size of this study was relatively small, and only correlation analysis between variables was performed. Correlation analysis is a cross-sectional method and cannot be used to infer causal relationships between variables. Thus, the relationships between AB, PSMU, and negative emotions should be further explored using other techniques. Fifth, most of the stimuli used in the two tasks are related more to platform names rather than specific types of content or behavioral engagements within platforms, which arguably are the things driving behavioral use and therefore problematic use. Finally, this study was a behavioral experiment, and the results may be more convincing if combined with eye-tracking technology or brain imaging.

In future studies, first, a variety of approaches including structured clinical interviews are needed to validate the diagnosis whether individuals are prone to PUMU. Second, we can decrease the stimulus presentation time to further explore the AB effect in the Stroop task. Third, SM-related images can be used as materials to improve the ecological impact of the stimuli. The Stroop task is a measure of an individual’s selective attention and response suppression, while DPT is a measure of an individual’s spatial attention. Other research paradigms such as flicker-induced change blindness paradigms, dual-task procedures, and visual search paradigms can be used to explore AB in PSMU. Fourth, the sample size can be increased to explore whether PSMU is a mediator or moderator between AB and negative emotions. Longitudinal tracking studies can be performed to explore the causal relationships between AB, PSMU, and negative emotions. Fifth, in terms of stimulus selection, specific types of content or behavioral engagements within SM platforms can be selected, rather than limited to SM-related names, so as to more objectively evaluate the AB of problematic users of SM. Lastly, the mechanism of AB in problematic users of SM can be explored by combining event-related potential analysis, eye-movement tracking, and brain imaging technology.

## Figures and Tables

**Figure 1 ijerph-19-16938-f001:**
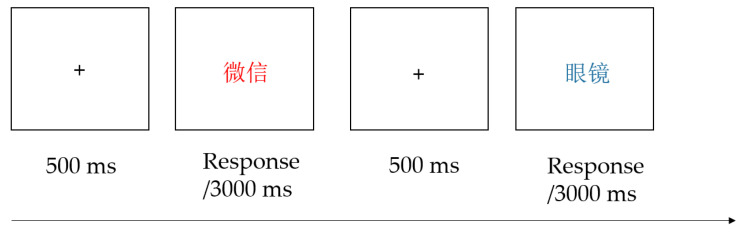
The timeline of the addiction Stroop task. *Note:* 微信 = WeChat, 眼镜 = glasses. Index of AB = mean RT of SM-related stimuli − mean RT of neutral stimuli.

**Figure 2 ijerph-19-16938-f002:**
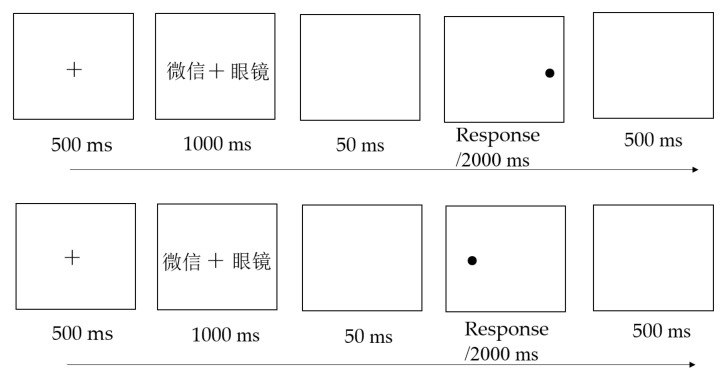
The timeline of the addiction DPT. *Note:* 微信 = WeChat, 眼镜 = glasses. Index of AB = mean RT of congruent trials − mean RT of incongruent trials.

**Figure 3 ijerph-19-16938-f003:**
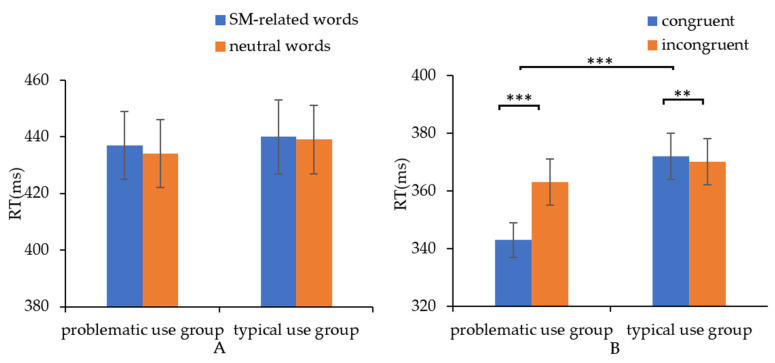
The mean RT of all the conditions in the Stroop task and DPT. (**A**) No significant difference was found between the problematic use group and the typical use group in the Stroop task. (**B**) Problematic users reacted faster to the congruent condition than to the incongruent condition, while typical users reacted more slowly to the congruent condition than to the incongruent condition in the DPT. ** *p* < 0.01, *** *p* < 0.001.

**Figure 4 ijerph-19-16938-f004:**
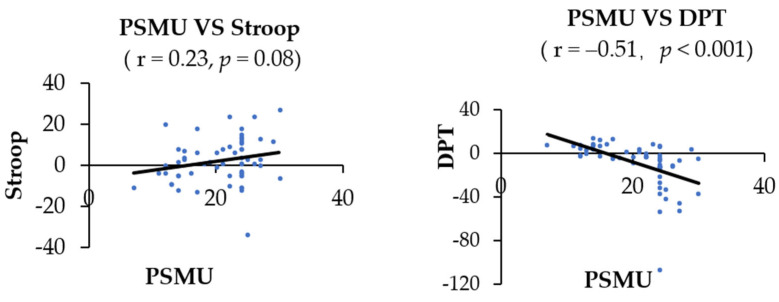
Correlation between PSMU and the index of AB in the addiction Stroop task and DPT. PSMU was negatively correlated with the index of AB in the addiction DPT (r = −0.51, *p* < 0.001).

**Figure 5 ijerph-19-16938-f005:**
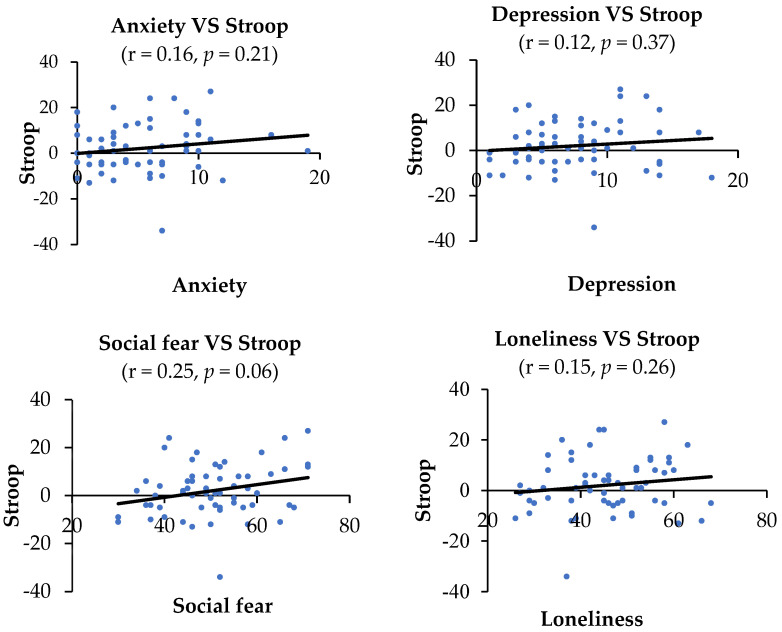
Correlations between negative emotions and the index of AB in the addiction Stroop and DPT. The index of AB was negatively correlated with anxiety (r = −0.37, *p* < 0.01), depression (r = −0.51, *p* < 0.001), and social fear (r = −0.30, *p* < 0.05) in the DPT.

**Table 1 ijerph-19-16938-t001:** Descriptive statistics of the problematic use group and typical use group (M ± SD).

	Problematic Use Group (*n* = 30)	Typical Use Group (*n* = 30)
BSMAS	24.90 ± 2.09	16.17 ± 4.15
Age (years)	20.07 ± 1.70	19.33 ± 1.40
Gender (% female)	73.33	67.67

*Note*: BSMAS = Bergen Social Media Addiction Scale.

**Table 2 ijerph-19-16938-t002:** Descriptive data and inter-correlations between variables (*n* = 60).

	M (SD)	1	2	3	4	5
1. BSMAS	20.62 (5.51)	1	0.28 *	0.35 **	0.38 **	0.19
2. Anxiety	5.23 (4.18)		1	0.72 ***	0.38 **	0.40 **
3. Depression	7.55 (4.04)			1	0.31 *	0.57 ***
4. Social fear	50.48 (10.27)				1	0.49 ***
5. Loneliness	45.47 (10.59)					1

*Note:* BSMAS = Bergen Social Media Addiction Scale. * *p* < 0.05, ** *p* < 0.01, *** *p* < 0.001.

## Data Availability

Data are available from the corresponding author upon reasonable request and with permission of School of Education in Guangzhou University.

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
