# Peer review of "Attentional Bias Is Associated with Negative Emotions in Problematic Users of Social Media as Measured by a Dot-Probe Task"

_ijerph, 2022, doi:10.3390/ijerph192416938_

Round 1

Reviewer 1 Report

Thank you for inviting me to review this paper. I was glad to see some inclusion of behavioural measures to help gain some additional insight into attentional bias/vigilance towards SM-related stimuli which I do believe provides some novel insight into this field. It is far more common to see manuscripts which more traditionally take questionable self-report scales and conduct cross-sectional analyses only. I have some observations which I feel need addressing however.

#1. The first statement sets up the assumption that “social media addiction” is an established behavioural addiction, but this is not accurate. There is no consensus within the academic community or representation in clinical manuals that this is recognised. My concern is that the authors do not appear to recognise the controversy of this issue and assume the position that SM “addiction” is an agreed and recognised concept/term. At the very least, I would urge the authors to include acknowledgement that there is debate in the community on the concept of SM addiction and to tone down the assumptions made. This might involve including more balanced evidence about the SM literature and some of the critical position papers on the issue of SM “addiction” or similar. Although the technology “addiction” field is well saturated there are wider concerns about the real value of measurements in this field and what they are actually measuring. A recent paper (see below) has demonstrated that you can take a meaningless concept (in this case “offline friend addiction” and produce a perfectly valid and reliable measure of this, based on using the typical protocols and analytic approaches common to this field. Further to this, there is growing discussion and concern about the theoretical basis for “tech addiction” and the fact that the field is becoming too quick to pathologise what is in most cases a very typical and normal everyday behavior (see Aagaard, 2020 below). This is an issue which applies to the current manuscript.

Aagaard, J. (2020). Beyond the rhetoric of tech addiction: why we should be discussing tech habits instead (and how). Phenomenology and the Cognitive Sciences. https://doi.org/10.1007/s11097-020-09669-z

Satchell, L., Fido, D., Harper, C., Shaw, H., Davidson, B. I., Ellis, D. A., Hart, C. M., Jalil, R., Jones, A., Kaye, L. K., Lancaster, G., & Pavetich, M. (2021). Development of an Offline-Friend Addiction Questionnaire (O-FAQ): Are most people really social addicts? Behavior Research Methods, 53, 1097–1106. https://doi.org/10.3758/s13428-020-01462-9

#2. There is a lack of detail about behaviours in relation to SM use constitute “addictive” use. “SM-related stimuli”. That is, the authors include an overview of the addiction component model and also theorise about the negative emotion correlates, but do not provide any specific insight into what types of behaviours or attention to specific types of content are hypothesised to explain “addictive” use. This is relevant as this would set up a much stronger basis for the selection of stimuli in the behavioural tasks. That is, the words seem to relate more to platform names rather than specific types of content or behavioural engagements within platforms, which arguably are the things driving behavioural use (and therefore “addictive” use). Some additional justification for the selection of the stimuli would be really helpful here and how these are theorised to be especially relevant to draw out vigilance effects for an “addicted” sample.

#3. Word stimuli for the Stroop task appeared to be controlled for similarity, frequency and length but not for other relevant features such as valence or how “social” they are. That is, a confound between experimental conditions is that there is a lack of control in level of valence of stimuli both between conditions and within them. Arguably, it might be expected that irrespective of severity of “addiction”, there is likely to be vigilance to stimuli with greater valence and towards stimuli with social information (e.g., “circle of friends”). I can’t help but think this is a mor confound to the experimental design and might at least in part explain any differences in the DPT effects.

#4. It is good the authors have distinguished between “addiction” and “typical” groups. However, when it comes to the correlations (which feature within the abstract as key findings), these should arguably be done by sub-group as well as overall, especially as the previous analysis suggests significant differences in SMA scores between these. This would be more meaningful to understand how SMA severity relates to negative affect variables. Similarly, this applies to section 3.4 when exploring correlations between index of AB with SMA scores. This should be done by sub-group rather than overall.

#5. “In the current study, we used self-report scales to study the influence of negative emotions on SMA.”- this statement is not accurate. The study was looking at relationships between SM and negative emotions”- analyses could not establish causal effects here.

Minor

#6. Might suggest using the word “typical” use rather than “normal use”

Author Response

Reply to Reviewer 1

Thank you for inviting me to review this paper. I was glad to see some inclusion of behavioural measures to help gain some additional insight into attentional bias/vigilance towards SM-related stimuli which I do believe provides some novel insight into this field. It is far more common to see manuscripts which more traditionally take questionable self-report scales and conduct cross-sectional analyses only. I have some observations which I feel need addressing however.

#1. The first statement sets up the assumption that “social media addiction” is an established behavioural addiction, but this is not accurate. There is no consensus within the academic community or representation in clinical manuals that this is recognised. My concern is that the authors do not appear to recognise the controversy of this issue and assume the position that SM “addiction” is an agreed and recognised concept/term. At the very least, I would urge the authors to include acknowledgement that there is debate in the community on the concept of SM addiction and to tone down the assumptions made. This might involve including more balanced evidence about the SM literature and some of the critical position papers on the issue of SM “addiction” or similar. Although the technology “addiction” field is well saturated there are wider concerns about the real value of measurements in this field and what they are actually measuring. A recent paper (see below) has demonstrated that you can take a meaningless concept (in this case “offline friend addiction” and produce a perfectly valid and reliable measure of this, based on using the typical protocols and analytic approaches common to this field. Further to this, there is growing discussion and concern about the theoretical basis for “tech addiction” and the fact that the field is becoming too quick to pathologise what is in most cases a very typical and normal everyday behavior (see Aagaard, 2020 below). This is an issue which applies to the current manuscript.

Aagaard, J. (2020). Beyond the rhetoric of tech addiction: why we should be discussing tech habits instead (and how). Phenomenology and the Cognitive Sciences. https://doi.org/10.1007/s11097-020-09669-z

Satchell, L., Fido, D., Harper, C., Shaw, H., Davidson, B. I., Ellis, D. A., Hart, C. M., Jalil, R., Jones, A., Kaye, L. K., Lancaster, G., & Pavetich, M. (2021). Development of an Offline-Friend Addiction Questionnaire (O-FAQ): Are most people really social addicts? Behavior Research Methods, 53, 1097–1106. https://doi.org/10.3758/s13428-020-01462-9

Reply: Thanks for the suggestion. Social media addiction is not currently recognized in either the Diagnostic and Statistical Manual of Mental Disorders, fifth edition, (DSM-V) or the International Classification of Diseases, 11th edition. The use of the word "addiction" may be potentially discriminating and stigmatizing. We have reviewed some literature and more literature defines such behavior as problematic social media use (PSMU) or excessive use of social media. Therefore, we suggest that it is better to use PUSM to replace the term “social media addiction”.

#2. There is a lack of detail about behaviours in relation to SM use constitute “addictive” use. “SM-related stimuli”. That is, the authors include an overview of the addiction component model and also theorise about the negative emotion correlates, but do not provide any specific insight into what types of behaviours or attention to specific types of content are hypothesised to explain “addictive” use. This is relevant as this would set up a much stronger basis for the selection of stimuli in the behavioural tasks. That is, the words seem to relate more to platform names rather than specific types of content or behavioural engagements within platforms, which arguably are the things driving behavioural use (and therefore “addictive” use). Some additional justification for the selection of the stimuli would be really helpful here and how these are theorised to be especially relevant to draw out vigilance effects for an “addicted” sample.

Reply: Thanks for the suggestion. More details about behaviours in relation to SM use constitute “addictive” use have been added in the revised manuscript (Lines 35-43). Most of the stimuli used in the two tasks are related more to platform names rather than specific types of content or behavioural engagements within platforms, which arguably are the things driving behavioural use (and therefore “addictive” use). This point is not fully considered in this study, which is supplemented in the research limitations (Lines 826-846).

#3. Word stimuli for the Stroop task appeared to be controlled for similarity, frequency and length but not for other relevant features such as valence or how “social” they are. That is, a confound between experimental conditions is that there is a lack of control in level of valence of stimuli both between conditions and within them. Arguably, it might be expected that irrespective of severity of “addiction”, there is likely to be vigilance to stimuli with greater valence and towards stimuli with social information (e.g., “circle of friends”). I can’t help but think this is a mor confound to the experimental design and might at least in part explain any differences in the DPT effects.

Reply: Thanks for the question. In the addiction Stroop task, 15 college students who did not participate in this experiment were asked to rate the extent of SM-related words and neutral words on a 5-point Likert scale (from 1 = extremely inconsistent to 5 = extremely consistent). Ultimately, 20 SM-related words (e.g. WeChat, Weibo, circle of friends) and 20 neutral words referring to daily necessities (e.g. desk, glasses, shampoo) were chosen for use in the formal experiment. Social media-related words and neutral words were matched in similarity, frequency, and length, respectively. In the addiction dot-probe task, stimuli were selected as described for the addiction Stroop task. In both tasks, the word stimulus was the same, including 20 social media-related words and 20 daily necessities as non-social media-related words. The difference was that in the Stroop task, the stimulus was presented in the center of the screen, while in the dot-probe task, the social media-related stimulus and neutral stimulus were paired and presented at both ends of the screen. Therefore, we do not think that the effect of DPT is due to different levels of valence of stimuli.

#4. It is good the authors have distinguished between “addiction” and “typical” groups. However, when it comes to the correlations (which feature within the abstract as key findings), these should arguably be done by sub-group as well as overall, especially as the previous analysis suggests significant differences in SMA scores between these. This would be more meaningful to understand how SMA severity relates to negative affect variables. Similarly, this applies to section 3.4 when exploring correlations between index of AB with SMA scores. This should be done by sub-group rather than overall.

Reply: Thanks for the suggestion. When analyzing the relationship between SMA and negative emotions, we tried to analyze the correlation between two sub-groups. It may be because there were too few data in each group (n=30), and there was no significant correlation between SMA and negative emotions in the addiction group. Also, the cut-off value of the Bergen Social Media Addiction Scale is 24 points, and scores of BSMAS in the problematic use group are very concentrated, with an average (25.07±1.86). The value span is small, and the correlation with other variables is not significant is understandable. Therefore, we analyzed the overall data in our analysis. In addition, a previous study has done the correlations by overall rather than sub-group (see Jiang, Zhao, & Li, 2017 below).

Jiang, Z., Zhao, X., & Li, C. (2017). Self-control predicts attentional bias assessed by online shopping-related Stroop in high online shopping addiction tendency college students. Compr Psychiatry, 75, 14-21. doi:10.1016/j.comppsych.2017.02.007

#5. “In the current study, we used self-report scales to study the influence of negative emotions on SMA.”- this statement is not accurate. The study was looking at relationships between SM and negative emotions”- analyses could not establish causal effects here.

Reply: Thanks for the suggestion. We have revised the statement in the revised manuscript. In the current study, we used self-report scales to look at relationships between PSMU and negative emotions (Lines 613-614).

#6. Might suggest using the word “typical” use rather than “normal use”

Reply: Thanks for the suggestion. We have used the word “typical use” rather than “normal use” in the revised manuscript.

Reviewer 2 Report

I believe that this study has clinically important implications for behavioral addiction research. But I have several questions to authors.

#1. Social media addiction can have various coexisting psychiatric problems such as depression, anxiety, and alcohol problems. I wonder if there was enough consideration for this. In particular, in Table 1, comparison of variables between the two groups was made only for the SMA scale, age, and gender.

#2. Only one scale appears to be used in diagnosing SNS addiction. Because SNS addiction is a controversial diagnosis, multiple methods should have been used to validate the diagnosis.

#3. I wonder if it has not been verified through structured clinical interviews, etc., whether the subjects have other coexisting problems. Overall, further clarification is needed in the selection and exclusion of subjects.

#4. It seems that the types of SNS used by the subjects varied. Diversity in types and patterns of SNS usage by subjects should have been considered. In addition, it should be evaluated whether other behavioral addiction problems, such as gaming addiction, did not coexist.

#5. It is doubtful whether presenting only words without using pictures would have been salient to the subjects. Is there any further evidence that word stimuli were sufficient?

#6. It seems that there are cases of confusion as to whether the presented word stimulus is related to social media or general social activity. For example, isn't the ‘circle of friends’ interpreted as a stimulus related to general social activity?

#7. The difference between the dot-probe task and the STROOP task is explained only in the presentation of stimuli. Is there any difference in the cognitive domain evaluated by the dot-probe task and the STROOP task?

Author Response

Reply to Reviewer 2

I believe that this study has clinically important implications for behavioral addiction research. But I have several questions to authors.

#1. Social media addiction can have various coexisting psychiatric problems such as depression, anxiety, and alcohol problems. I wonder if there was enough consideration for this. In particular, in Table 1, comparison of variables between the two groups was made only for the SMA scale, age, and gender.

Reply: Thanks for the suggestion. Social media addiction can have various coexisting psychiatric problems. When recruiting participants, inclusion criteria included fluency in Chinese, normal or corrected-to-normal vision, and the use of SM. Exclusion criteria included current or previous psychiatric conditions (schizophrenia, mood disorders, obsessive compulsive disorder and somatoform disorders) and addiction behaviors other than social media addiction. In the study, individuals with depression, anxiety and alcohol problems were excluded. In addition, the demographic factors associated with social media addiction include age and gender. Therefore, when screening participants in this study, the two sub-groups showed differences in scores of the Bergen Social Media Addiction Scale, while there was no significant difference in age and gender. A similar study can be seen in the previous study, in which only age was controlled for demographic factors of the two sub-group (see Hu et al., 2020 below).

Hu, Y., Guo, J., Jou, M., Zhou, S., Wang, D., Maguire, P., . . . Qu, F. (2020). Investigating the attentional bias and information processing mechanism of mobile phone addicts towards emotional information. Computers in Human Behavior, 110. doi:10.1016/j.chb.2020.106378

#2. Only one scale appears to be used in diagnosing SNS addiction. Because SNS addiction is a controversial diagnosis, multiple methods should have been used to validate the diagnosis.

Reply: Thanks for the suggestion. In view of the fact that the Bergen Social Media Addiction Scale is the most commonly used tool in the research on social media addiction behaviors, this study only used this scale to distinguish whether individuals have addiction tendencies. However, at present, the diagnosis of this behavior is controversial, and multiple methods should be adopted to distinguish this behavior, which is not fully considered in this study. Therefore, for the above point, we made a supplement in the limitations of the study (Lines 800-802).

#3. I wonder if it has not been verified through structured clinical interviews, etc., whether the subjects have other coexisting problems. Overall, further clarification is needed in the selection and exclusion of subjects.

Reply: Thanks for the question. In this study, structured clinical interviews were used to determine whether participants had current or previous psychiatric conditions (schizophrenia, mood disorders, obsessive compulsive disorder and somatoform disorders) and addiction behaviors other than PSMU, and participants who did not meet the inclusion criteria were excluded. Further clarification in the selection and exclusion of subjects has been added in the revised manuscript (Lines 282-288).

#4. It seems that the types of SNS used by the subjects varied. Diversity in types and patterns of SNS usage by subjects should have been considered. In addition, it should be evaluated whether other behavioral addiction problems, such as gaming addiction, did not coexist.

Reply: Thanks for the suggestion. The types of SNS used by the subjects varied and diversity in types and patterns of SNS usage by subjects have been considered in the study. When selecting social media-related stimuli, this study selects common characteristic stimuli of various social media in China, such as login, forwarding, etc. In addition, we have supplemented the inclusion criteria and the exclusion criteria in the selection of participants (Lines 282-288).

#5. It is doubtful whether presenting only words without using pictures would have been salient to the subjects. Is there any further evidence that word stimuli were sufficient?

Reply: Thanks for the suggestion. So far, there is no evidence that word stimuli were sufficient. In this study, word stimuli were used in both the Stroop task and the DPT as experimental materials, and the ecological effect of text may not be as good as that of pictures. This is one of the limitations of this study and we have added it in the limitation section of the study. In future studies, SM-related images can be used as materials to improve the ecological impact of the study (Lines 804-806).

#6. It seems that there are cases of confusion as to whether the presented word stimulus is related to social media or general social activity. For example, isn't the ‘circle of friends’ interpreted as a stimulus related to general social activity?

Reply: Thanks for the question. Wechat is the most widely used social media platform in Chinese college students. The ‘circle of friends’ is a social function launched by Wechat, which is a small and private circle built by acquaintances. Users can follow and share their friends’ lives in circle of friends, so as to strengthen the connection with others. Before starting the experiment, it was necessary to select to social media relevant words and neutral words as stimulus materials. The valence of social media related words and neutral words were evaluated by college students who did not participate in this experiment, and the results show that circle of friends is highly correlated with social media and has a relatively good valence.

#7. The difference between the dot-probe task and the STROOP task is explained only in the presentation of stimuli. Is there any difference in the cognitive domain evaluated by the dot-probe task and the STROOP task?

Reply: Thanks for the suggestion. The addiction Stroop and dot-probe task are two commonly used measures of attentional bias which inferred through participants’ reaction time performance when a addiction-related stimulus is presented. The Stroop task is more a measure of an individual’s selective attention and response suppression. The dot-probe task is a measure of an individual's spatial attention. The difference between the dot-probe task and the Stroop task has been added in the revised manuscript (Lines 837-838).

Round 2

Reviewer 1 Report

Authors have addressed previously identified comments

Reviewer 2 Report

Appropriate responses and corrections have been made to my questions.